# Trait anxiety and depressive rumination mediate the effect of perceived childhood rearing on adulthood presenteeism

**Akifumi Shimasaki[1,2], Ayaka Deguchi[1], Yoshitaka Ishii[1], Tomoteru Seki[1], Yoshio Iwata[1], Yu Tamada[3], Mina Honyashiki****[1], Yota Fujimura[3], Takeshi Inoue****[1]\*, Jiro Masuya[1]**

**1** Department of Psychiatry, Tokyo Medical University, Shinjuku-ku, Tokyo, Japan, **2** Department of Psychiatry, Maruyamasou Hospital, Ishioka, Ibaraki, Japan, **3** Department of Psychiatry, Tokyo Medical University Hachioji Medical Center, Tokyo, Japan

\* tinoue@tokyo-med.ac.jp

## Abstract

### Aim

Productivity loss in the workplace due to physical or mental health problems, which is called presenteeism, leads to large financial losses. Personal and work environment factors, as well as physical and mental illnesses are associated with presenteeism, but the detailed underlying mechanism remains unclear. In this study, we analyzed the effects of perceived childhood rearing on the presenteeism of adult workers, and the mediating effects of trait anxiety and depressive rumination.

### Methods

In 2017 and 2018, a cross-sectional paper-based survey was conducted, and written consent from 447 adult workers was obtained. Demographic information and results from the Parental Bonding Instrument, trait anxiety of State-Trait Anxiety Inventory Form Y, Ruminative Responses Scale, and Work Limitations Questionnaire were surveyed. Multiple regression analyses and structural equation modeling were conducted.

### Results

The low perceived quality of rearing from parents in childhood, i.e., low care and high overprotection, indirectly worsened current presenteeism via trait anxiety and depressive rumination. Presenteeism was directly worsened by trait anxiety and depressive rumination, and the low perceived quality of rearing from parents directly affected trait anxiety and depressive rumination, and trait anxiety affected depressive rumination.

### Conclusion

The results of this study clarified the long-term influences of the low perceived quality of rearing experienced in childhood on adulthood presenteeism via trait anxiety and depressive rumination. Therefore, assessing the quality of childhood rearing, trait anxiety, and

Data are available from the Internal Review Board of the Department of Psychiatry, Tokyo Medical University, Japan (contact via email: seisinka@tokyo-med.ac.jp) for researchers who meet the criteria for access to confidential data. A minimal data set is provided as Supplementary Data S1.

**Funding:** This work was partly supported by a Grant-in-Aid for Scientific Research (no. 21K07510, to TI) from the Japanese Ministry of Education, Culture, Sports, Science and Technology (https://www.jsps.go.jp/english/egrants/). The funders had no role in the study design, data collection and analysis, decision to publish, or preparation of the manuscript.

**Competing interests:** I have read the journal's policy and the authors of this manuscript have the following competing interests: Yota Fujimura has received personal compensation from Sumitomo Pharma, and grants from Otsuka Pharmaceutical, Sumitomo Pharma, and Shionogi. Takeshi Inoue has received personal compensation from Mochida Pharmaceutical, Takeda Pharmaceutical, Eli Lilly, Janssen Pharmaceutical, MSD, Taisho Toyama Pharmaceutical, Yoshitomiyakuhin, and Daiichi Sankyo; grants from Shionogi, Astellas, Tsumura, and Eisai; and grants and personal compensation from Otsuka Pharmaceutical, Sumitomo Pharma, Mitsubishi Tanabe Pharma, Kyowa Pharmaceutical Industry, Pfizer, Novartis Pharma, and Meiji Seika Pharma; and is a member of the advisory boards of Pfizer, Novartis Pharma, and Mitsubishi Tanabe Pharma. Jiro Masuya has received personal compensation from Otsuka Pharmaceutical, Eli Lilly, Astellas, and Meiji Yasuda Mental Health Foundation, and grants from Pfizer. All other authors declare that they have no actual or potential conflicts of interest associated with this study. This does not alter our adherence to PLOS ONE policies on sharing data and materials.

depressive rumination of individuals may help to elucidate the causes of presenteeism in the workplace, and how to manage it effectively.

## Introduction

Various stressors have been identified as risk factors that affect common mental health disorders, such as depressive disorders, as well as general mental health. Childhood experiences of abuse, inadequate parental child-rearing, bullying, and unhappy experiences have long-term effects on mental health in adulthood [1–4]. Among them, negative parental child-rearing experiences are well-known to be risk factors for anxiety disorders, suicidality, neuroticism, depression, and other psychiatric disorders [1, 5–7]. Child-rearing received from parents reportedly influences occupational stress via its impact on neuroticism [8]. Occupational stress strongly affects productivity loss (presenteeism) in the workplace, and presenteeism in the workplace is a major socioeconomic problem for countries as well as individuals [9–11]. Reduced worker productivity due to illness, i.e., presenteeism, is a significant concern in workplaces, and has a high social cost [11]. Presenteeism is also a major challenge in the field of health care; improving presenteeism can be a goal when aiming for patient recovery. Identifying risk factors for presenteeism may lead to the development of new medical or social policies. The experience of negative parental rearing in childhood, as well as abuse and bullying in childhood, was recently reported to have long-term effects on presenteeism [12–15],

Depressive rumination is persistently thinking negatively about one's own depressed symptoms, and the causes and results of depression [16]. Depressive rumination is a risk factor for depression [16–18], and depression itself can cause occupational loss, such as presenteeism as well as absenteeism [11, 19]. Given the known close association between depression and presenteeism [19], it is presumed that depressive rumination also affects presenteeism, but this has not been confirmed to date. On the other hand, childhood maltreatment enhances depressive symptoms and dysphoria by enhancing depressive rumination [20–23]. In addition, the Habit development, EXecutive control, Abstract processing, GOal discrepancies, and Negative bias (H-EX-A-GON) model [24], which is an integrated cognitive and psychological model for the onset and maintenance of depressive rumination, notes that a variety of biological and psychological factors, as well as unhelpful and overcontrolling rearing styles affect depressive rumination. A number of previous studies have also clarified the association between perceived rearing from parents in childhood and depressive rumination, and a 3-way chain association among perceived childhood rearing, depressive rumination, and presenteeism has been postulated, but to our knowledge, this has never been confirmed previously.

Recently, we performed a covariance structure analysis that demonstrated associations among 4 factors: childhood maltreatment affects trait anxiety, which in turn affects depressive rumination and enhances adulthood depressive symptoms [21]. In other words, trait anxiety is a mediating factor in the effects of childhood maltreatment on depressive rumination, and the results suggested that high trait anxiety due to adverse childhood experiences may exist in the background of depressive rumination. Several previous studies have proposed the mediation theory that personality traits, such as trait anxiety and neuroticism, are mediating factors in the effects of adverse childhood experiences on adult mental health [5, 6, 15, 21, 25]. Trait anxiety mediates the association between childhood maltreatment and presenteeism in adulthood [15]. It is possible that trait anxiety may act as a mediating factor in the above-mentioned 3-way association among rearing from parents in childhood, depressive rumination, and presenteeism. On the other hand, rearing from parents in childhood has also been reported to significantly

affect trait anxiety [26]. Therefore, a new 4-factor chain association in which rearing from parents in childhood influences trait anxiety, which in turn influences presenteeism through its effect on depressive rumination, has been considered but has not been confirmed to date.

Based on the results of previous studies and the theoretical basis described above, in this study we built the hypothesis that perceived childhood rearing influences trait anxiety, that these preparatory factors further influence depressive rumination, and that perceived childhood rearing and trait anxiety influence presenteeism through their influence on depressive rumination. We assessed perceived rearing in childhood, trait anxiety, depressive rumination, and presenteeism in adult volunteers using a paper-based questionnaire survey, and analyzed the associations and mediating effects among these factors using structural equation modeling.

## Methods

### Participants

From 2017 to 2018, self-report paper-based questionnaires were distributed to 1,237 adult volunteers from the community, who were recruited by convenience sampling through our acquaintances at Tokyo Medical University. The study was part of a larger study [8, 9, 13]. A total of 447 workers (196 men and 251 women; average age, 40.9 ± 11.8 years), who gave written informed consent for the academic use of their information, were included in the analysis. The inclusion criterion was being 20 years or older and the exclusion criterion was having a severe physical or organic brain disease. Participating in this study was voluntary, and nonparticipation led to no disadvantage. The subjects were informed that their personal information would be treated anonymously and would remain confidential. In accordance with the Declaration of Helsinki (amended in Fortaleza in 2013), this study was conducted under the approval of the Ethics Committee of Tokyo Medical University (study approval no.: SH3502).

### Questionnaires

**Work Limitations Questionnaire (WLQ) Short Form.**  The WLQ is a self-report questionnaire to evaluate the health-associated disability to work. Its short form consists of 4 subscales: each includes 2 items evaluated on a 6-point scale (1 to 6) [27]. This study used its Japanese short version, which has been validated previously [28]. The percentage of the loss of productivity owing to health problems in the previous 2 weeks can be evaluated by this measurement. A high %productivity loss means low work performance, i.e., high presenteeism.

**Trait anxiety subscale of State-Trait Anxiety Inventory Form Y (STAI-Y).**  The STAI-Y is a self-report questionnaire for the evaluation of trait anxiety and state anxiety [29]. This study used the subscale of trait anxiety, which is a relatively stable tendency to react to anxiety-evoking events. Twenty items for trait anxiety were evaluated on a 4-point scale (1 to 4). This study used its Japanese version, which has been validated previously [30].

**Parental Bonding Instrument (PBI).**  The PBI is a retrospective evaluation of parental rearing attitudes experienced in childhood [31]. Its Japanese version was developed and validated previously [32]. Twenty-five items of the PBI are divided to 2 subscales, i.e., care (12 items), and overprotection (13 items), and each item is assessed using a 4-point scale (0 to 3). A high score for "care" indicates that a subject received more appropriate rearing from his/her parents (more emotional warmth, more empathy, more closeness, less neglect, and less indifference). A high score for "overprotection" indicates that a subject received more overprotective rearing from his/her parents (more overcontrol, more intrusion, more excessive contact, more infantilization, and prevention of independent behavior and autonomy). This study used the total scores of each subscale for the father and the mother for analysis.

**Ruminative Responses Scale (RRS).** The RRS is a self-report evaluation of the frequency of depressive rumination, and consists of 22 items, assessed using a 4-point scale (1 to 4) [33]. This study used its Japanese version, which has been validated previously [34]. The total score of the 22 items was analyzed.

## Statistical analysis

Statistical analysis of the association of demographic information and questionnaire data was performed using SPSS Statistics Version 28 (IBM, Armonk, NY, USA).

The structural equation modeling was analyzed by Mplus 8.5 software (Muthén & Muthén, Los Angeles, CA, USA): covariance structure analysis was used with the robust maximum likelihood estimation method. All coefficients were standardized. The root mean square error of approximation (RMSEA) and the comparative fit index (CFI) were used for goodness-of-fit: RMSEA < 0.08 and CFI > 0.95 were considered to indicate an acceptable model fit, and RMSEA < 0.05 and CFI > 0.97 were considered to indicate a good model fit [35]. A $p$-value of less than 0.05 was considered to indicate statistical significance. In this study, a structural equation model of perceived childhood rearing to directly affect trait anxiety and depressive rumination, and to indirectly affect presenteeism through trait anxiety and depressive rumination was built.

## Results

### Demographic characteristics of the subjects and their association with presenteeism

Table 1 shows the demographic characteristics and questionnaire data of the 447 adult workers. Unmarried status, past history of psychiatric disease, and current psychiatric disease were associated with high presenteeism (WLQ %productivity loss). Paternal care, paternal overprotection, maternal care, and maternal overprotection of the PBI, trait anxiety of the STAI-Y, and depressive rumination of the RRS were significantly associated with presenteeism. Other demographic factors were not associated with presenteeism.

### Multiple regression analysis using the forced entry method

Multiple regression analysis was conducted with presenteeism (WLQ %productivity loss) as the dependent variable, and the other variables as independent variables (Table 2). Current psychiatric disease, education years, RRS depressive rumination score, and STAI-Y trait anxiety score were identified as factors significantly associated with presenteeism. Among these significant variables, RRS depressive rumination score, and STAI-Y trait anxiety score showed the highest significance. The coefficient of determination (adjusted $R^2$) of this model on presenteeism was 0.220; i.e., this model explains 22.0% of the variability of presenteeism.

### Structural equation model including rearing from parents, trait anxiety, depressive rumination, and presenteeism

In this study, a structural equation model was built as shown in Figs 1 and 2, with perceived childhood rearing (PBI), trait anxiety (STAI-Y), depressive rumination (RRS), and % productivity loss of the WLQ (presenteeism). Regarding the direct effects in the structural equation model of parental overprotection (Fig 1A), parental overprotection directly increased trait anxiety and depressive rumination. In addition, trait anxiety directly increased depressive rumination, and trait anxiety and depressive rumination directly worsened presenteeism. However, the direct effect of parental overprotection on presenteeism was not statistically significant. The

**Table 1. Demographic and clinical characteristics and questionnaire measures of 447 adult workers, and their correlation with %productivity loss of the WLQ.**

| Characteristic or measure | Value (% of subjects or mean ± SD) | Correlation with %productivity loss of the WLQ or its comparison between categories |
|---|---|---|
| Age (years) | 40.9 ± 11.8 | $r = -0.060$, $p = 0.209$ (n.s.) |
| Sex (men: women) | 43.8%: 56.2% | Men: 4.1 ± 4.2<br>Women: 4.4± 4.2, $p = 0.514$ (n.s.) |
| Education (years) | 14.8 ± 1.8 | $r = -0.092$, $p = 0.052$ (n.s.) |
| Marital status (married: unmarried) | 64.9%: 35.1% | Married: 3.7 ± 4.0<br>Unmarried: 5.3 ± 4.4, $p < 0.001$*** |
| Past history of psychiatric disease (yes: no) | 11.2%: 88.8% | Yes: 5.8 ± 4.6<br>No: 4.1 ± 4.1, $p = 0.006$** |
| Current psychiatric disease (yes: no) | 4.3%: 95.7% | Yes: 7.8 ± 5.0<br>No: 4.1 ± 4.0, $p < 0.001$*** |
| Family history of psychiatric disease (yes: no) | 11.8%: 88.2% | Yes: 4.2 ± 3.7<br>No: 4.2 ± 4.3, $p = 0.926$ (n.s.) |
| PBI-paternal care | 23.8 ± 8.0 | $r = -0.129$, $p = 0.006$** |
| PBI-paternal overprotection | 9.6 ± 6.8 | $r = 0.206$, $p < 0.001$*** |
| PBI-maternal care | 28.3 ± 6.8 | $r = -0.159$, $p < 0.001$*** |
| PBI-maternal overprotection | 9.7 ± 6.9 | $r = 0.199$, $p < 0.001$*** |
| STAI-Y trait anxiety | 43.2 ± 10.5 | $r = 0.420$, $p < 0.001$*** |
| RRS rumination score | 35.3 ± 11.2 | $r = 0.375$, $p < 0.001$*** |
| WLQ %productivity loss | 4.2 ± 4.2 | |

Data are presented as means ± standard deviation (SD) or numbers. $r$ = Pearson correlation coefficient; n.s., not significant

** $p < 0.01$

*** $p < 0.001$

PBI, Parental Bonding Instrument; STAI-Y, State-Trait Anxiety Inventory form Y; RRS, Ruminative Responses Scale; WLQ, Work Limitations Questionnaire

coefficient of determination ($R^2$) of presenteeism of this model was 0.218; i.e., this model explains 21.8% of the variability of presenteeism. Significant indirect effects were the paths from parental overprotection to depressive rumination via trait anxiety, from parental overprotection to presenteeism via trait anxiety, from parental overprotection to presenteeism via depressive rumination, from parental overprotection to presenteeism via trait anxiety and depressive rumination, and from trait anxiety to presenteeism via depressive rumination (Fig 1B).

Regarding the direct effects in a structural equation model of parental care (Fig 2A), parental care directly decreased trait anxiety and depressive rumination. In addition, trait anxiety directly increased depressive rumination, and trait anxiety and depressive rumination directly worsened %productivity loss of the WLQ (presenteeism). However, the direct effect of parental care on presenteeism was not statistically significant. The coefficient of determination ($R^2$) of presenteeism of this model was 0.209; i.e., this model explains 20.9% of the variability of presenteeism. Significant indirect effects were the paths from parental care to depressive rumination via trait anxiety, from parental care to presenteeism via trait anxiety, from parental care to presenteeism via trait anxiety and depressive rumination, and from trait anxiety to presenteeism via depressive rumination (Fig 2B). The path from parental care to presenteeism via depressive rumination was not statistically significant.

Subsequently, to test the contribution of the father and the mother separately, we constructed 2 additional structural equation models. Each structural equation model incorporated

**Table 2. Results of multiple regression analysis of %productivity loss of the WLQ.**

| Independent variable | Standardized partial regression coefficient (β) | $p$-value | VIF |
|---|---|---|---|
| PBI paternal care | 0.050 | 0.390 | 1.912 |
| PBI paternal overprotection | 0.066 | 0.308 | 2.349 |
| PBI maternal care | 0.024 | 0.705 | 2.321 |
| PBI maternal overprotection | 0.049 | 0.461 | 2.467 |
| STAI-Y trait anxiety | 0.282 | $< 0.001$ | 1.568 |
| RRS rumination | 0.184 | $< 0.001$ | 1.624 |
| Age (years) | −0.044 | 0.383 | 1.396 |
| Sex (men = 1; women = 2) | −0.031 | 0.480 | 1.102 |
| Education (years) | −0.100 | 0.045 | 1.381 |
| Marital status (married = 2; unmarried = 1) | −0.062 | 0.188 | 1.253 |
| Current psychiatric disease (yes = 2; no = 1) | 0.103 | 0.038 | 1.370 |
| Past history of psychiatric disease (yes = 2; no = 1) | −0.012 | 0.806 | 1.383 |

Adjusted $R^2$ = 0.220, $F$ = 11.28, $p < 0.001$

VIF, variance inflation factor; PBI, Parental Bonding Instrument; STAI-Y, State-Trait Anxiety Inventory form Y; RRS, Ruminative Responses Scale; WLQ, Work Limitations Questionnaire

the latent variables of either 'maternal parenting' or 'paternal parenting', which were composed of observed care and overprotection variables (namely, the care and overprotection subscores for the mother and those for the father). The structural equation model incorporating 'maternal parenting' is shown in S1 Fig. 1 (S1 Appendix), and the model incorporating 'paternal parenting' is shown in S1 Fig. 2 (S1 Appendix). The results obtained from these 2 additional structural equation models suggest that the contribution of the father's parenting to trait anxiety, depressive rumination, and presenteeism is almost similar to the contribution of the

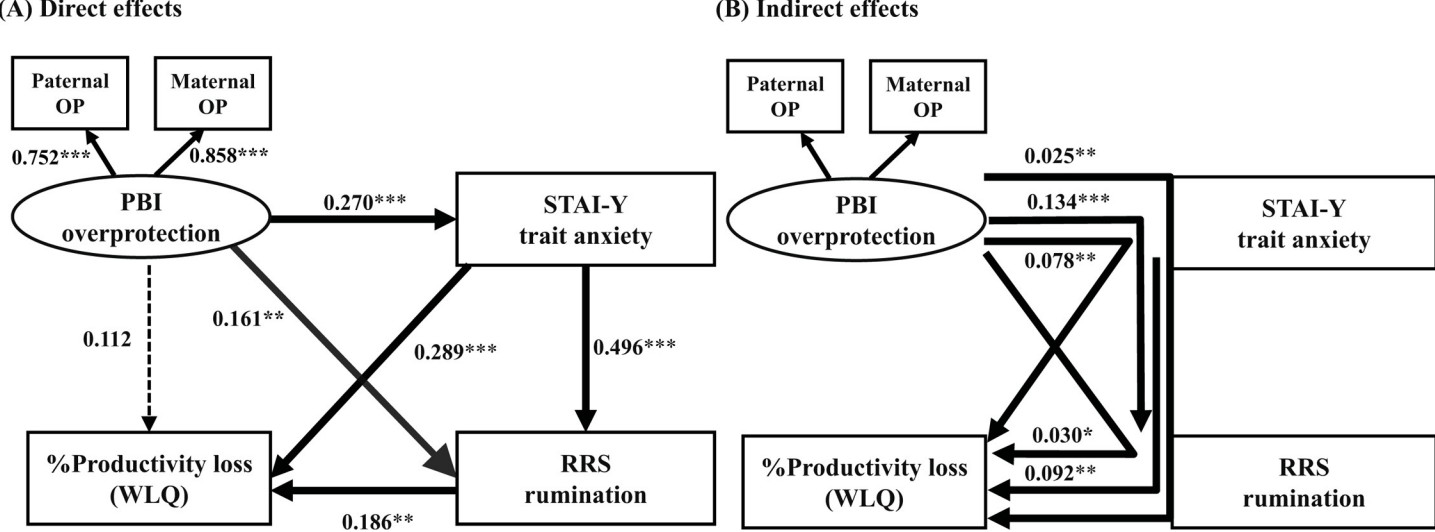

**Fig 1. Results of the structural equation model with "overprotection (OP)" of the PBI as the latent variable, and trait anxiety (STAI-Y), depressive rumination (RRS), and %productivity loss (presenteeism) on the WLQ as the observed variables.** The latent variable is shown as an oval, and the observed variables are shown as rectangles. Direct effects (A) and indirect effects (B) between the variables are shown. The numbers show the standardized path coefficients. $^*p < 0.05$, $^{**}p < 0.01$, $^{***}p < 0.001$.

**(A) Direct effects**          **(B) Indirect effects**

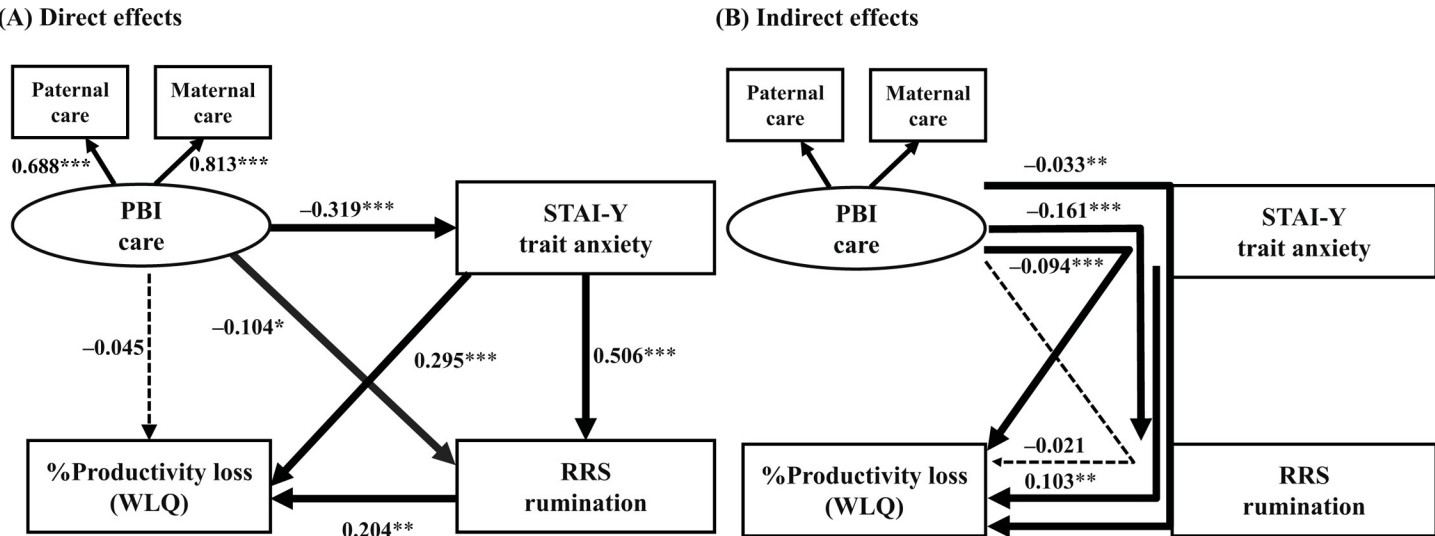

**Fig 2. Results of the structural equation model with "care" of the PBI as the latent variable, and trait anxiety (STAI-Y), depressive rumination (RRS), and % productivity loss (presenteeism) on the WLQ as the observed variables.** The latent variable is shown as an oval, and the observed variables are shown as rectangles. Direct effects (A) and indirect effects (B) between the variables are shown. The numbers show the standardized path coefficients. *$p < 0.05$, **$p < 0.01$, ***$p < 0.001$.

mother's parenting. However, the direct effect of the father's parenting on depressive rumination was cancelled in the additional structural equation model.

## Discussion

This study demonstrated through structural equation modeling that perceived rearing from parents in childhood affects presenteeism through its effects on trait anxiety and depressive rumination. The Parental Bonding Instrument used to assess perceived rearing from parents in childhood evaluates 2 subscales of rearing, namely, parental care and overprotection, both of which have opposite effects on trait anxiety, depressive rumination, and presenteeism. Although the effects of perceived rearing from parents in childhood on presenteeism have been reported [14], the association between these 2 factors mediated by trait anxiety and depressive rumination has not been previously reported, and to our knowledge, the present study is the first to report such an association. Previous studies from our research group reported that trait anxiety and depressive rumination mediate the effects of childhood abuse on depressive symptoms [21, 36], and that trait anxiety mediates the effects of childhood abuse on presenteeism [15]. As childhood abuse and depressive symptoms in the previous studies were considered to be associated with inadequate rearing from parents and presenteeism, respectively [1, 19], these previous studies partially support the validity of the results of the present study. Regarding factors that contribute to presenteeism in the workplace, trait anxiety and depressive rumination are considered to be personal factors that work together to influence presenteeism, and furthermore, perceived rearing from parents in childhood is a more remote personal factor that influences presenteeism.

Overcontrolling by parents during childhood reportedly correlates with depressive rumination in college students [37]. Furthermore, prospective studies of adolescent subjects also confirmed that inadequate rearing from parents (e.g., expression of submissive negative emotions, and lack of positive support and feedback from mothers) enhanced rumination in the children. [38, 39] Thus, the association between perceived rearing from parents experienced in childhood and depressive rumination has been reported in several previous studies, and depressive

rumination was identified as a mediating factor in the effects of rearing form parents on the development of depression [37]. The results of the present study on the association between perceived rearing from parents experienced in childhood and depressive rumination are consistent with the findings of several previous studies. The clinical question that then arises is, why does inadequate childhood rearing from parents enhance depressive rumination? The results of the present study suggest that inadequate childhood rearing from parents may influence children's personality traits, and increase their susceptibility to anxiety, which in turn may lead to higher rumination. Of course, it is also possible that inadequate childhood rearing from parents enhances depressive rumination, which in turn enhances anxiety symptoms, but in any case, it is clear that the 3 factors are synchronously variable. The findings of our recent report that childhood abuse, including neglect, enhances depressive rumination through its effects on trait anxiety [21] are also partially consistent with the results of the present study.

As mentioned in the Introduction section, low productivity in the workplace, i.e., presenteeism, is a major socioeconomic problem for countries as well as individuals [9–11]. The main contributors to presenteeism are various diseases, including common mental health problems, such as anxiety and depressive disorders [10]. Not only illnesses, but also workplace stress and work environment affect presenteeism [40–42]. Our research group has recently reported that personal factors, such as childhood victimization, childhood abuse, quality of parental care experienced in childhood, neuroticism, insomnia, and dietary habits also influence presenteeism [9, 13, 14, 43, 44]. Because of the long time interval between the experience of rearing from the parents in childhood and the status of mental health in adulthood, the presence of some long-lasting mediating factors is expected to be essential for such an association. Previous studies identified subjective cognitive function and neuroticism as mediators of the effects of rearing from parents in childhood and childhood victimization experiences on presenteeism, respectively [13, 14]. The present study further clarified the psychological mechanism of presenteeism by identifying trait anxiety and depressive rumination as important individual factors associated with presenteeism, and as mediators of the effect of perceived rearing from parents in childhood on presenteeism. In particular, the mediating effects of trait anxiety and depressive rumination were full and not partial, because the direct effect of perceived rearing from parents in childhood on presenteeism was not significant in our structural equation model. Thus, our results suggest that it is essential to focus on both factors when considering the effects of perceived rearing from parents in childhood on presenteeism.

The results of our present study indicate that inappropriate rearing from parents experienced in childhood, and the resulting trait anxiety and depressive rumination are the causes of low worker productivity, i.e., presenteeism, in the workplace. As rearing from parents experienced in childhood cannot be modified in adulthood, interventions for trait anxiety and depressive rumination are effective in improving presenteeism; selective serotonin reuptake inhibitors, the transcendental meditation technique, and exercise therapy have been reported to attenuate trait anxiety [45–47]. Thus, these therapies have the potential to improve presenteeism, but this needs to be tested in clinical trials. Cognitive behavioral therapy focused on depressive rumination, and mindfulness-based cognitive therapy were shown to be effective in lowering depressive rumination [24, 48]. Although it is not possible to retrospectively modify adults' past experiences of low-quality rearing from their parents, preventing low-quality care and overprotection in children through social interventions will help prevent future presenteeism when they become adults. Early intervention of presenteeism with the above psychological and social factors in mind would be beneficial for individuals and society in general, in terms of improving social costs.

This study has several limitations. This study was conducted using a questionnaire that relies on past memories, which may have caused recall bias. In addition, as the study was

conducted on adult volunteers, there are limitations in applying the results to patients with depression and other psychiatric disorders. Furthermore, as this was a cross-sectional study, a long-term prospective study is needed in the future to conclude the association among the low perceived quality of rearing experienced in childhood, trait anxiety, depressive rumination, and presenteeism in adulthood. In the present study, we hypothesized that trait anxiety influences depressive rumination, because trait anxiety is a relatively stable response to anxiety-inducing experiences and regarded as a personality trait. However, the reverse pathway, in which higher rumination influences higher trait anxiety, is also possible, and needs to be analyzed prospectively in the future. As not only overprotection and low quality care, as analyzed in this study, but also various other factors, such as abuse, bullying, and harassment also contribute to high trait anxiety, these adverse experiences may influence our model, and should be analyzed in the future.

## Conclusions

This study suggests that the quality of rearing (overprotection and low care) received from parents in childhood influence presenteeism in the workplace in adulthood, and that trait anxiety and depressive rumination fully mediate these associations. Our results suggest that when addressing the presenteeism of individual workers, it is necessary to focus on their perceived rearing from parents in childhood, trait anxiety, and depressive rumination as personal factors involved in presenteeism.

## Supporting information

**S1 Appendix. Supplementary figures.**
(PDF)

**S1 Data. Minimal data set.**
(CSV)

## Acknowledgments

We thank Dr. Nobutada Takahashi of Fuji Psychosomatic Rehabilitation Institute Hospital, Dr. Hiroshi Matsuda of Kashiwazaki Kosei Hospital, Dr. Yasuhiko Takita (deceased) of Maruyamasou Hospital, and Dr. Yoshihide Takaesu of Izumi Hospital for their collection of subject data. We thank Dr. Helena Popiel of the Center for International Education and Research, Tokyo Medical University, for editorial review of the manuscript.

## Author Contributions

**Conceptualization:** Akifumi Shimasaki, Takeshi Inoue.

**Data curation:** Akifumi Shimasaki, Takeshi Inoue, Jiro Masuya.

**Formal analysis:** Akifumi Shimasaki, Takeshi Inoue.

**Funding acquisition:** Takeshi Inoue.

**Investigation:** Akifumi Shimasaki, Takeshi Inoue.

**Methodology:** Akifumi Shimasaki, Takeshi Inoue.

**Project administration:** Akifumi Shimasaki, Takeshi Inoue, Jiro Masuya.

**Supervision:** Akifumi Shimasaki, Takeshi Inoue, Jiro Masuya.

**Validation:** Akifumi Shimasaki, Ayaka Deguchi, Yoshitaka Ishii, Tomoteru Seki, Yoshio Iwata, Yu Tamada, Mina Honyashiki, Yota Fujimura, Takeshi Inoue, Jiro Masuya.

**Visualization:** Akifumi Shimasaki.

**Writing – original draft:** Akifumi Shimasaki, Ayaka Deguchi, Yoshitaka Ishii, Tomoteru Seki, Yoshio Iwata, Yu Tamada, Mina Honyashiki, Yota Fujimura, Takeshi Inoue, Jiro Masuya.

**Writing – review & editing:** Akifumi Shimasaki, Ayaka Deguchi, Yoshitaka Ishii, Tomoteru Seki, Yoshio Iwata, Yu Tamada, Mina Honyashiki, Yota Fujimura, Takeshi Inoue, Jiro Masuya.

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
