## [Decision Letter · Decision Letter 0]

24 May 2023

PONE-D-23-04844Trait anxiety and depressive rumination mediate the effect of perceived childhood rearing on adulthood presenteeismPLOS ONE

Dear Dr. Inoue,

Thank you for submitting your manuscript to PLOS ONE. After careful consideration, we feel that it has merit but does not fully meet PLOS ONE’s publication criteria as it currently stands. Therefore, we invite you to submit a revised version of the manuscript that addresses the points raised during the review process.

We look forward to receiving your revised manuscript.

Kind regards,

Juan-Luis Castillo-Navarrete, Ph.D.

Academic Editor

PLOS ONE

Journal Requirements:

"This work was partly supported by a Grant-in-Aid for Scientific Research (no. 21K07510, to TI) from the Japanese Ministry of Education, Culture, Sports, Science and Technology (https://kaken.nii.ac.jp/en/)."

"I have read the journal’s policy and the authors of this manuscript have the following competing interests: Yota Fujimura has received personal compensation from Sumitomo Pharma, and grants from Otsuka Pharmaceutical, Sumitomo Pharma, and Shionogi. Takeshi Inoue has received personal compensation from Mochida Pharmaceutical, Takeda Pharmaceutical, Eli Lilly, Janssen Pharmaceutical, MSD, Taisho Toyama Pharmaceutical, Yoshitomiyakuhin, and Daiichi Sankyo; grants from Shionogi, Astellas, Tsumura, and Eisai; and grants and personal compensation from Otsuka Pharmaceutical, Sumitomo Pharma, Mitsubishi Tanabe Pharma, Kyowa Pharmaceutical Industry, Pfizer, Novartis Pharma, and Meiji Seika Pharma; and is a member of the advisory boards of Pfizer, Novartis Pharma, and Mitsubishi Tanabe Pharma. Jiro Masuya has received personal compensation from Otsuka Pharmaceutical, Eli Lilly, Astellas, and Meiji Yasuda Mental Health Foundation, and grants from Pfizer. All other authors declare that they have no actual or potential conflicts of interest associated with this study. This does not alter our adherence to PLOS ONE policies on sharing data and materials."

We note that you received funding from a commercial source: Sumitomo Pharma, Otsuka Pharmaceutical, Shionogi, Mochida Pharmaceutical, Takeda Pharmaceutical, Eli Lilly, Janssen Pharmaceutical, MSD, Taisho Toyama Pharmaceutical, Yoshitomiyakuhin, Daiichi Sankyo, Astellas, Tsumura, Eisai, Mitsubishi Tanabe Pharma, Kyowa Pharmaceutical Industry, Pfizer, Novartis Pharma, Meiji Seika Pharma and Meiji Yasuda Mental Health Foundation.

Within this Competing Interests Statement, please confirm that this does not alter your adherence to all PLOS ONE policies on sharing data and materials by including the following statement: "This does not alter our adherence to PLOS ONE policies on sharing data and materials.” (as detailed online in our guide for authors http://journals.plos.org/plosone/s/competing-interests).  

If there are restrictions on sharing of data and/or materials, please state these. Please note that we cannot proceed with consideration of your article until this information has been declared. 

6. We note you have included a table to which you do not refer in the text of your manuscript. Please ensure that you refer to Table 2 in your text; if accepted, production will need this reference to link the reader to the Table.

**Additional Editor Comments:**

Given such an interesting subject, the writing is of great quality, so the indications of our reviewers have been minimal, so I have no doubt that it will not be a major problem to consider them.

Reviewers' comments:

Reviewer's Responses to Questions

**Comments to the Author**

1. Is the manuscript technically sound, and do the data support the conclusions?

Reviewer #1: Yes

Reviewer #2: Yes

2. Has the statistical analysis been performed appropriately and rigorously? 

Reviewer #1: Yes

Reviewer #2: Yes

3. Have the authors made all data underlying the findings in their manuscript fully available?

Reviewer #1: Yes

Reviewer #2: Yes

4. Is the manuscript presented in an intelligible fashion and written in standard English?

Reviewer #1: Yes

Reviewer #2: Yes

5. Review Comments to the Author

Reviewer #1: In general terms, it seems to me that it is a very interesting work that raises a novel idea that visualizes parenthood in other spectrums beyond its link with child development.

Between lines 60 and 64 of the paper, where reference is made to depression, it seems to me that the idea of the loss of occupationality that depression itself generates in people should be highlighted (to reinforce the idea that is being developed in the same paragraph).

In the part of the results in Tables 1 and 2, why is the difference between father and mother not made in the STAI-Y trait anxiety and RRS rumination score variables? It would be interesting to see how the productivity loss variable impacts on both subsamples separately. Having the multiple regression analysis, is there any variable that has a greater influence or dominance over the others in presenteeism?

Reviewer #2: Studiyng possible causes of presenteeism is no doubt of interest and its strog association with trait axiety and depressive rumiation seems out of dicussion. Nevertheless the hypothesis that one influnces the other and both are related to percieved rearing from parents in childhood requires further studies with a longitudinal design. Other limitation are wrihtly aknowleged by the authors.

6. PLOS authors have the option to publish the peer review history of their article (what does this mean?). If published, this will include your full peer review and any attached files.

Reviewer #1: No

Reviewer #2: No

---

## [Author Response · Author response to Decision Letter 0]

1 Jul 2023

POINT-BY-POINT RESPONSES TO THE REVIEWERS’ COMMENTS

We thank the reviewers for the helpful advice and comments, which have enabled us to substantially improve our manuscript.

Journal Requirements:

Response:

Thank you for the comment. We rechecked our manuscript and formatted it to be in accordance with the style requirements of PLOS ONE, including the file naming.

Response:

We would like to change ‘Funding Information’ to the following. We would be grateful if you could please change the online submission form.

“Funding Information

This work was partly supported by a Grant-in-Aid for Scientific Research (no. 21K07510, to TI) from the Japanese Ministry of Education, Culture, Sports, Science and Technology (https://www.jsps.go.jp/english/egrants/). The funders had no role in the study design, data collection and analysis, decision to publish, or preparation of the manuscript.”

3. Thank you for stating the following in the Competing Interests section. Within this Competing Interests Statement, please confirm that this does not alter your adherence to all PLOS ONE policies on sharing data and materials by including the following statement: "This does not alter our adherence to PLOS ONE policies on sharing data and materials.”

Response:

Our original submission already included the statement “This does not alter our adherence to PLOS ONE policies on sharing data and materials.” in the ‘Competing Interests’ section. Therefore, we do not think any change is necessary, but we apologize if we have not understood your explanation correctly.

4. In your Data Availability statement, you have not specified where the minimal data set underlying the results described in your manuscript can be found. PLOS defines a study's minimal data set as the underlying data used to reach the conclusions drawn in the manuscript and any additional data required to replicate the reported study findings in their entirety.

Response:

Thank you for your comment. We would like to change ‘Data Availability Statement’ to the following and add a minimal data set (a csv file) as Supplementary Data S1. We would be grateful if you could please change the online submission form.

“Data Availability Statement: Data cannot be shared publicly because of Ethics Committee restriction. All relevant data are within the paper. Data are available from the Internal Review Board of the Department of Psychiatry, Tokyo Medical University, Japan (contact via email: seisinka@tokyo-med.ac.jp) for researchers who meet the criteria for access to confidential data. A minimal data set is provided as Supplementary Data S1.” 

5. We note you have included a table to which you do not refer in the text of your manuscript. Please ensure that you refer to Table 2 in your text; if accepted, production will need this reference to link the reader to the Table.

Response:

We apologize that we did not refer to Table 2 in our original manuscript. We have referred to Table 2 in our revised manuscript (page 11, line 189).

Response:

We confirmed that all references are correct and that we have not included any retracted papers.

REVIEWER #1

1. Between lines 60 and 64 of the paper, where reference is made to depression, it seems to me that the idea of the loss of occupationality that depression itself generates in people should be highlighted (to reinforce the idea that is being developed in the same paragraph).

Response: 

We thank you for your comments. In accordance with your suggestion, we added a description that depression itself results in occupational loss, such as presenteeism as well as absenteeism, as follows.

“Depressive rumination is a risk factor for depression [16-18], and depression itself can cause occupational loss, such as presenteeism as well as absenteeism [11,19].” (page 4, lines 61 to 63)

2. In the part of the results in Tables 1 and 2, why is the difference between father and mother not made in the STAIY trait anxiety and RRS rumination score variables? It would be interesting to see how the productivity loss variable impacts on both subsamples separately. Having the multiple regression analysis, is there any variable that has a greater influence or dominance over the others in presenteeism?

Response:

Thank you for your suggestions. A major reason that there are only small differences between paternal and maternal parenting in the effect on trait anxiety and depressive rumination is that all these parenting variables are correlated (r = 0.414–0.645). 

 In accordance with your suggestion, to test the contribution of the father and mother separately, we constructed 2 additional structural equation models for presenteeism. Each structural equation model incorporated the latent variables of either ‘paternal parenting’ or ‘maternal parenting’, which were composed of observed care and overprotection variables (namely, the care and overprotection subscores for the father, and the care and overprotection subscores for the mother). The results obtained from the 2 additional structural equation models suggest that the contribution of the father’s parenting for trait anxiety, depressive rumination, and presenteeism is similar to that of the mother’s parenting, although the direct effect from the father’s parenting to depressive rumination was cancelled (S1 Figs 1 and 2). These results were added to the Results section, as follows, and as Supplementary S1 Figs 1 and 2.

“Subsequently, to test the contribution of the father and the mother separately, we constructed 2 additional structural equation models. Each structural equation model incorporated the latent variables of either ‘maternal parenting’ or ‘paternal parenting’, which were composed of observed care and overprotection variables (namely, the care and overprotection subscores for the mother and those for the father). The structural equation model incorporating ‘maternal parenting’ is shown in Supplementary S1 Fig. 1, and the model incorporating ‘paternal parenting’ is shown in Supplementary S1 Fig. 2. The results obtained from these 2 additional structural equation models suggest that the contribution of the father’s parenting to trait anxiety, depressive rumination, and presenteeism is almost similar to the contribution of the mother’s parenting. However, the direct effect of the father’s parenting on depressive rumination was cancelled in the additional structural equation model.” (page 13, line 229 to page 14, line 239)

In Table 2 of the original manuscript, the multiple regression analysis showed that trait anxiety and depressive rumination had substantial effects on presenteeism, and these factors mediated the effect of parenting in the structural equation model. To make this point clear, we added the following description to the revised manuscript.

“Among these significant variables, RRS depressive rumination score, and STAI-Y trait anxiety score showed the highest significance.” (page 12, lines 191 to 192)

 

REVIEWER #2

1. Studying possible causes of presenteeism is no doubt of interest and its strong association with trait anxiety and depressive rumination seems out of discussion. Nevertheless the hypothesis that one influences the other and both are related to perceived rearing from parents in childhood requires further studies with a longitudinal design. Other limitation are rightly acknowledged by the authors.

Response:

We thank you for your comments. To verify our hypothesis, we would like to further study with a longitudinal design.

---

## [Editor Report · Decision Letter 1]

21 Jul 2023

Trait anxiety and depressive rumination mediate the effect of perceived childhood rearing on adulthood presenteeism

PONE-D-23-04844R1

Dear Dr. Takeshi Inoue,

We’re pleased to inform you that your manuscript has been judged scientifically suitable for publication and will be formally accepted for publication once it meets all outstanding technical requirements.

Kind regards,

Juan-Luis Castillo-Navarrete, Ph.D.

Academic Editor

PLOS ONE

Additional Editor Comments (optional):

The study is a great contribution to the subject, with a very good approach and that reflects a great work, receive my congratulations.
---

## [Editor Report · Acceptance letter]

26 Jul 2023

PONE-D-23-04844R1 

Trait anxiety and depressive rumination mediate the effect of perceived childhood rearing on adulthood presenteeism 

Dear Dr. Inoue:

I'm pleased to inform you that your manuscript has been deemed suitable for publication in PLOS ONE. Congratulations! Your manuscript is now with our production department. 

Kind regards, 

on behalf of

Dr. Juan-Luis Castillo-Navarrete 

Academic Editor

PLOS ONE